# φ-OTDR Based on Orthogonal Frequency-Division Multiplexing Time Sequence Pulse Modulation

Zhengyang Li, Yangan Zhang *, Xueguang Yuan, Zhenyu Xiao, Yuan Zhang and Yongqing Huang

School of Electronic Engineering, Beijing University of Posts and Telecommunications, Beijing 100876, China;
lizhengyang@bupt.edu.cn (Z.L.); yuanxg@bupt.edu.cn (X.Y.); zyxiao@bupt.edu.cn (Z.X.);
zhang_yuan@bupt.edu.cn (Y.Z.); yqhuang@bupt.edu.cn (Y.H.)
* Correspondence: zhang@bupt.edu.cn

**Abstract:** This study introduces an innovative phase-sensitive optical time-domain reflectometer (φ-OTDR) technology based on orthogonal frequency-division multiplexing (OFDM) and nonlinear frequency modulation (NLFM) pulse modulation sequences. The proposed approach addresses the inherent trade-offs among spatial resolution, frequency response range, and sensing distance that conventional φ-OTDR systems encounter. This method optimizes spatial resolution and sensing distance by modulating both the frequency and phase of optical pulses. Moreover, it enhances sidelobe suppression by adjusting the nonlinearity of frequency modulation, reducing interference between adjacent signals, and improving the signal-to-noise ratio (SNR). Additionally, orthogonal frequency-division multiplexing expands the frequency response range. This paper elucidates the fundamental principles and implementation of OFDM-NLFM time-domain pulse modulation techniques and designs, experimentally validates a φ-OTDR system based on this method, and conducts comprehensive testing and analysis of the system's performance. The experimental results demonstrate that the proposed φ-OTDR system achieves an 11 m spatial resolution and a frequency response range of 1–10 kHz over a 16.3 km optical fiber, utilizing a 65 MHz frequency bandwidth with multiplexed signals across four frequencies. This innovative approach reduces hardware resource consumption, opening up promising prospects for various practical engineering applications in optical fiber sensing technology.

**Keywords:** optical fiber sensing; phase-sensitive optical time-domain reflectometer; orthogonal frequency-division multiplexing; nonlinear frequency modulation



## 1. Introduction

Fiber optic sensing technology is a crucial engineering technique that has demonstrated widespread applications in various fields, including industrial automation, perimeter security, aerospace, and more. Among these applications, phase-sensitive OTDR, also known as φ-OTDR, has gained significant popularity in the current landscape of fiber optic sensing technology. This is due to its numerous advantages, such as high spatial resolution, extended sensing distances, and immunity to the effects of fiber connections [1–4].

The φ-OTDR system's frequency response range is inherently constrained by sensing distance, where greater distances yield narrower attainable bandwidths [5–7]. Current methods to widen this bandwidth include positive and negative pulse signals, periodic non-uniform sampling, and frequency division multiplexing (FDM). Notably, in 2017, D. Chen et al. introduced a DAS system employing FDM-TGD-OFDR, achieving a 9 kHz frequency response across a 24.7 km distance with modulation pulses spanning five frequency segments within a 100 MHz bandwidth [8]. Similarly, in 2018, J. Xiong et al. combined positive and negative pulse signals, FDM technology, and chirped pulse φ-OTDR, achieving a spatial resolution of 9.3 m and a frequency range of 0–24 kHz over a 2.1 km sensing distance using 195 MHz modulation bandwidth and four-frequency segment

FDM [9]. Moreover, in 2023, Z. Xiao et al. proposed an equivalent sampling method based on compressed sensing and interval-scanning pulses, thus enhancing the frequency response of phase-sensitive optical time-domain reflectometry through periodic non-uniform sampling [10]. Among these approaches, FDM emerged as a particularly effective approach in extending the frequency response range. However, this necessitates protective frequency bands to prevent interference between different frequency segments, thus demanding substantial modulation bandwidth and high-bandwidth detectors for achieving high-frequency response bandwidths [11–13]. The spatial resolution of the φ-OTDR system relies on pulse width, and achieving higher resolution necessitates narrower pulses. However, this reduction in pulse width decreases the injected optical energy, leading to a lower SNR and reduced sensing distances [14–16]. Consequently, conventional φ-OTDR systems confront a trade-off between spatial resolution and sensing distance, which poses challenges in meeting both high-resolution and long-distance sensing requirements [17,18]. Among the available solutions, linear frequency modulation (LFM) signals often feature narrow main lobe widths but lower sidelobe suppression ratios. To enhance sidelobe suppression, time-domain windowing is commonly employed. Nevertheless, this time-domain approach can diminish the main lobe's height, resulting in reduced spatial resolution [19–21]. Hence, a shift in frequency modulation techniques becomes imperative to improve the sidelobe suppression ratio. Differing from some other frequency modulation techniques, such as linear frequency modulation (LFM), the frequency modulation characteristics of non-linear frequency modulation (NLFM) signals contribute to reducing sidelobes in the time-domain signal. This enhances the prominence of the main lobe and improves the signal-to-noise ratio, thus enhancing the performance of optical fiber sensing systems. This is particularly crucial in optical fiber sensing applications where better signal quality is required. In 2019, J. Zhang et al. achieved a sensing range of 80 km and a spatial resolution of 2.7 m by employing iterative pre-distortion in combination with non-linear frequency modulation techniques. However, the frequency response range of this system was limited to 10–610 Hz [22]. In 2022, Y. Muanenda et al. utilized the readily adaptable direct digital synthesis (DDS) of pulses technique to generate compressed pulses with NLFM waveforms, thus achieving a spatial resolution of 0.5 m at a distance of 1.13 km. Nonetheless, this system required a spectrum resource of 500 MHz [23]. Hence, it is evident that the incorporation of NLFM technology alone with φ-OTDR systems still necessitates addressing concerns related to frequency response range and hardware resource utilization.

This paper introduces an φ-OTDR technique that combines OFDM and NLFM time-series pulse modulation. It employs multiple orthogonal subcarriers to efficiently stack spectra, enhancing spectrum utilization and extending the system's frequency response range. Within the finite detector bandwidth, this approach allows for more frequency band multiplexing. NLFM technology is used to fine-tune frequency modulation nonlinearity, resulting in reduced sidelobe levels, improved sidelobe suppression ratios, reduced interference between neighboring signals, and an enhanced SNR. The combination of OFDM and NLFM in pulsed light generates wider pulse widths, enabling more optical energy injection into the sensing fiber and yielding higher backscattered optical power in long-distance sensing scenarios. Consequently, this approach enhances spatial resolution and SNR, extending sensing distances. The experimental results demonstrate that utilizing OFDM-NLFM time-series pulse modulation with a 65 MHz bandwidth achieves an 11 m spatial resolution and a 1–10 kHz frequency response range over a 16.3 km optical fiber by multiplexing four frequencies. This research has the goal of improving fiber optic sensing technology to make it more practical for applications in industries such as industrial automation, security, and aerospace. This study's objectives include expanding the frequency response range of φ-OTDR systems, enhancing spatial resolution, and extending sensing distances. The introduction of novel modulation techniques is intended to tackle the limitations of current technology, thereby driving progress in fiber optic sensing, increasing its utility, and fostering its utilization in engineering applications and scientific research.

## 2. Principle and Theoretical Analysis

### 2.1. System Principles

The schematic diagram of the φ-OTDR based on OFDM-NLFM time-domain pulse modulation is shown in Figure 1. It utilizes an OFDM-NLFM time-domain signal as the pulse modulation signal. This sequence consists of $N$ pulses, and $N$ detection pulses are multiplexed within one sequence period $T$. The pulse repetition period is $T/N$, where the sequence period $T$ is greater than the maximum round-trip time of light for detection pulses through the long-distance sensing fiber. The value of $N$ depends on the maximum response frequency to be achieved and the ratio determined by the sensing distance for sequence repetition frequency. The pulse width of each detection pulse is $\tau_p(\tau_p < T/N)$, where $\tau_p$ determines the energy of the detection pulse light, and this energy cannot exceed the threshold of fiber nonlinearity effects. The detection pulse is an NLFM signal with a modulation bandwidth of $B_S$, which dictates the system's spatial resolution. The frequencies of multiple detection pulses are orthogonal to each other. Additionally, $f_0$ represents the central frequency of the detection pulse electrical signal, and the central frequency of each detection pulse varies with a step size of $\Delta f$. Through OFDM, the input signal is divided into multiple parallel signals, with each modulated on independent orthogonal subcarriers. This effectively utilizes the available spectrum and overcomes the bandwidth limitations in long-distance sensing. NLFM, with its non-linear variation of instantaneous frequency over time, enhances the sidelobe suppression ratio, reduces interference between adjacent signals, and improves the system's SNR [24–28]. The system combines the advantages of OFDM and NLFM, simultaneously enhancing spatial resolution, frequency response range, and sensing distance.

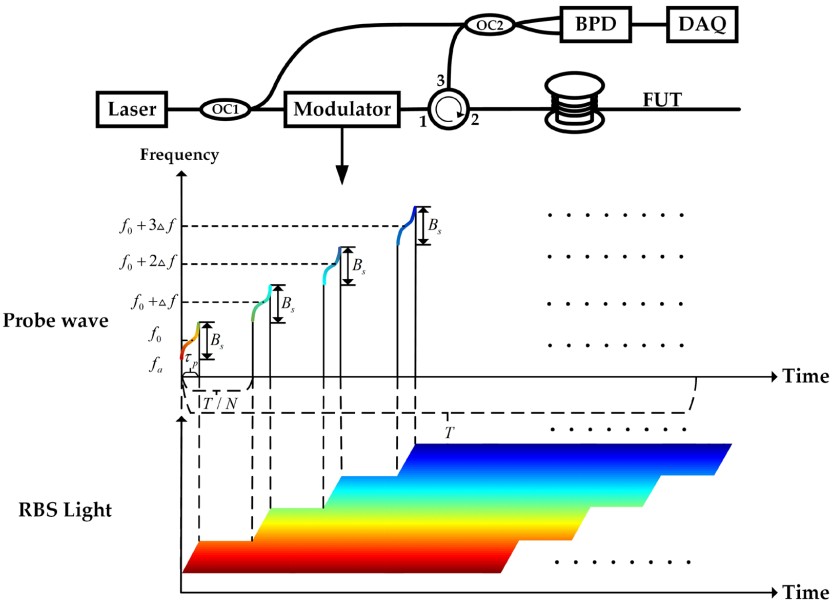

**Figure 1.** Schematic diagram of φ-OTDR based on OFDM-NLFM.

The probing light is modulated into chirp pulses, which can be represented as:

$$s(t) = rect(t/\tau_p)exp\{j2\pi f_a t + j\pi k t^2\} \tag{1}$$

where $rect(t/\tau_p)$ is a rectangular window function within the time range $t \in [0, \tau_p]$, $\tau_p$ is the pulse width, $f_a$ is the initial frequency of the chirp pulse signal, and $k$ is the frequency chirp rate [29].

After coherent detection and photodetection, the photocurrent signal is transformed into a complex signal through the Hilbert transformation in the digital domain. To maximize

the SNR, a matched filter should be used for data processing. In the digital domain, the matched filter $s^*(-t)$ is generated to process $i(t)$, resulting in the output signal:

$$r_c(t) = i(t) \otimes s^*(-t) = h(t) \otimes R(t) \tag{2}$$

Here, $\otimes$ denotes convolution, and $r_c(t)$ represents the trace line of the synthesized reflectivity of the test fiber. $R(t)$ can be considered as the target detection pulse. Through the matched filtering process, the original chirped frequency-modulated pulse $s(t)$ is compressed into $R(t)$. $R(t)$ is a function similar to sine and can be represented as:

$$R(t) = s(t) \otimes s^*(-t) = rect(\frac{t}{2\tau_p})\frac{sin[\pi kt(\tau_p - |t|)]}{\pi kt}exp\{-j2\pi(f_a + \frac{k\tau_p}{2})t\} \tag{3}$$

The system's spatial resolution $R(t)$ is determined by the full width at half maximum (FWHM) of the main lobe and is inversely proportional to the chirp frequency range, which is independent of the pulse width. This breaks the mutual constraint between spatial resolution and sensing distance. Therefore, by selecting appropriate chirped modulation pulses, higher spatial resolution can be achieved at longer sensing distances.

The matched filter is the optimal linear filter for white noise signals. If the double-sided power spectral density of white noise is $N_f$, then the SNR at time $t_0$ is given by

$$SNR = \frac{P_s}{P_n} = \frac{s_o(t_0)s_o^*(t_0)}{E[n_o(t)n_o^*(t)]} \leq \frac{E_s}{N_f} \tag{4}$$

In the equation, $P_s$ represents the power of the filtered output signal $s_o(t)$ at time $t_0$, $P_n$ is the average power of the filtered output noise $n_o(t)$, $E[\cdot]$ denotes the expectation function, and $E_s$ is the energy of the input signal $s_i(t)$. When matched filtering is employed, the SNR can reach its maximum value. It is evident from this formula that the achievable SNR of the system depends solely on the energy of the input waveform and is independent of other details, such as the modulation type of the input signal. Therefore, for systems with a constant white noise level, increasing the power and duration of the detection waveform is the only way to enhance the SNR.

The use of chirp frequency modulation can increase the effective linewidth of laser pulses, thereby reducing the Brillouin gain and raising the threshold for stimulated Brillouin scattering (SBS). Additionally, when the carrier spacing is greater than the Brillouin gain bandwidth, employing multi-carrier pulses distributes pulse energy across optical sideband carriers, and the SBS threshold is determined by the carrier with the highest spectral power, further reducing the Brillouin intensity. Therefore, utilizing chirp-modulated pulses and multi-carrier pulses enables the injection of higher-energy laser pulses into the sensing fiber, enhancing the SNR for long-distance sensing while also expanding the frequency response range.

## 2.2. Generation of NLFM Signals

In φ-OTDR systems, the detection signal primarily consists of the backscattered light from the pulsed optical signal, forming a continuous signal that varies with distance. Excessive sidelobes can lead to crosstalk between Rayleigh signals at different locations [30]. This paper employs NLFM instead of LFM to achieve lower sidelobe levels and enhance the system's SNR.

The theoretical foundation of NLFM signals is based on the phase progression principle [31]. NLFM signal models are diverse, and generating these signals is more complex compared to LFM signals, which lack precise signal design methods. In practice, various approximation methods are employed.

LFM signals are typically represented as

$$s(t) = u(t)e^{j2\pi f_0 t} = \frac{1}{\sqrt{\tau_p}}rect(\frac{t}{\tau_p})e^{j2\pi(f_0 t + \frac{1}{2}\mu t^2)} \tag{5}$$

$$u(t) = \frac{1}{\sqrt{\tau_p}} rect(\frac{t}{\tau_p}) e^{j2\pi(f_0 t + \frac{1}{2}\mu t^2)} \tag{6}$$

Assuming the signal bandwidth is $B$, then $\mu = B/\tau_p$ represents the frequency modulation rate. By taking the derivative of the phase of the signal $s(t)$, we obtain the instantaneous frequency of the signal:

$$F(t) = \frac{1}{2\pi} \frac{d(\pi\mu t^2 + 2\pi f_0 t)}{dt} = f_0 + \mu t \tag{7}$$

This paper employs the classical window function inversion method [32,33] to generate NLFM signals by altering the duration of different frequency components within the signal while maintaining constant instantaneous power. Once a window function $W$ is given, the group delay function $T$ of the waveform can be approximately solved using the stationary phase principle. Then, the phase function is determined based on the instantaneous frequency. The specific process of signal generation is as follows:

The NLFM signal is represented as:

$$x(t) = a(t)\exp(j\theta(t)) \tag{8}$$

In the equation, $a(t)$ represents the signal's amplitude, which is typically a constant value, and $\theta(t)$ represents the signal's phase function. Assuming the spectrum of $x(t)$ is $X(f)$, then the frequency response of its matched filter should be:

$$H(f) = X^*(f)e^{-j\Theta(f)} \tag{9}$$

In the equation, "*" represents complex conjugation, $\Theta(f) = 2\pi f t_0$, where $t_0$ is the moment at which the peak of the matched filter's output signal component is formed, and $e^{-j\Theta(f)}$ is the linear phase factor. In this case, the frequency spectrum $Y(f)$ of the matched filter's output $y(t)$ is given by:

$$Y(f) = \int_{-\infty}^{+\infty} y(t)\exp(-j2\pi f t)dt = X(f)H(f) = |X(f)|^2 \tag{10}$$

For a certain frequency-domain window function $W(f)$, let:

$$|X(f)|^2 = W(f) \tag{11}$$

According to the phase unwrapping principle, it is known that there is a relationship between the signal's spectrum and the frequency modulation rate, which is given by

$$X(f) \propto \sqrt{\frac{1}{\theta''(t)}} = \sqrt{\frac{1}{\frac{df(t)}{dt}}} = \sqrt{\frac{dT(f)}{df}} \tag{12}$$

In the equation, $T(f)$ represents the group delay function of the signal, and $f(t)$ is the frequency modulation function, which are inverse functions of each other. Assuming that the selected signal's spectrum after pulse compression is denoted as $W(f)$, then $W(f) \propto \frac{dT(f)}{df}$. Therefore, the group delay function can be obtained by integrating $W(f)$ as follows:

$$T(f) = K \int_{-\infty}^{f} W(v)dv \tag{13}$$

In the equation, *K* is a constant coefficient. When it is required that the NLFM signal has a bandwidth of *B* and a duration of $t_p$, we have:

$$K = \frac{t_p}{\int_{-B/2}^{B/2} W(v)dv} \tag{14}$$

Based on the relationship between $T(f)$ and $f(t)$, we can obtain:

$$f(t) = T^{-1}(f) \tag{15}$$

Therefore, the phase function of the signal is:

$$\theta(t) = 2\pi \int_{-\infty}^{t} f(\tau)d\tau \tag{16}$$

At this point, the generation of the NLFM signal $x(t)$ is complete.

The time–frequency spectrum of LFM is shown in Figure 2a, and the compressed pulse is shown in Figure 2b. Using the method described earlier, NLFM pulses with a sweep range of 20 MHz and a pulse width of 4 µs were generated through simulation. The compressed pulse's amplitude, after undergoing matched filtering, is shown in Figure 2b. In the figure, the main lobe width of the compressed pulse is measured at 10.2000 m, with a sidelobe suppression ratio reaching 12.8631 dB. In comparison to LFM, as shown in Figure 2b, where the main lobe width of LFM is 6.1850 and the sidelobe suppression ratio is 6.63492 dB, Figure 2c demonstrates the instantaneous frequency of NLFM, and Figure 2d shows the relative frequency spectra. It is evident that NLFM exhibits significant improvements in both main lobe width and sidelobe suppression ratio.

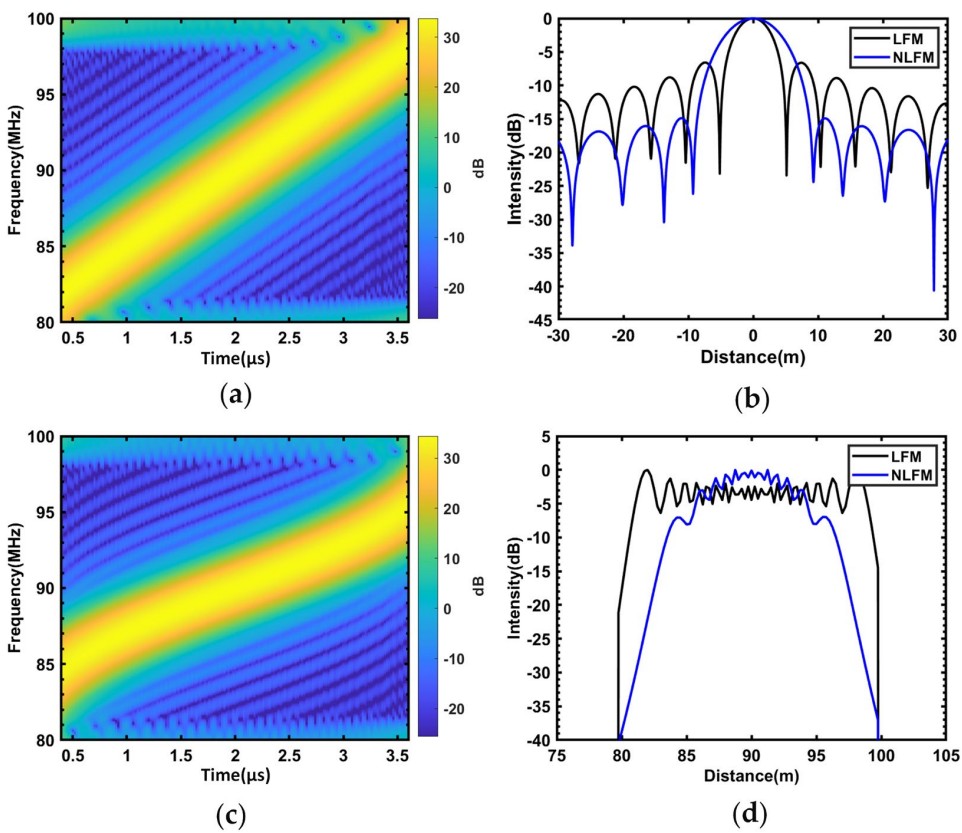

**Figure 2.** (**a**) Time-frequency spectra of LFM; (**b**) compressed pulses; (**c**) time-frequency spectra of NLFM; (**d**) relative frequency spectra of LFM and NLFM.

### 2.3. Generation of OFDM-NLFM Signals

When a pulse consisting of $M_t$ individual frequency-modulated pulses enters the sensing optical fiber, where each individual pulse has a duration of $\tau_p$ and a frequency modulation bandwidth of $B_s$, and the frequencies of adjacent individual pulses are separated by $\Delta f$, with the initial pulse having a central frequency of $f_0$, the central frequency of the signal for the $m$ pulse is represented as $f_m = f_0 + (m-1)\Delta f$. The bandwidth of all pulse signals is denoted as $B = B_s + (m-1)\Delta f$.

Assuming the $m$ waveform's pulse-compressed output spectrum is represented by the combined window spectrum $W_m(f)$, then

$$W_m(f) = \frac{\sum\limits_{i=1}^{N} a_i W_i(f)}{\sum\limits_{i=1}^{N} a_i} \tag{17}$$

According to Equation (12), the group delay $T_m(f)$ of the $m$ pulse signal can be obtained by integrating Equation (16).

The frequency modulation function of the $m$ pulse signal can be derived from the group delay as follows:

$$f_m(t) = T_m^{-1}(f) \tag{18}$$

Similarly, integrating Equation (17) yields the phase function:

$$\theta_m(t) = 2\pi \int_{-\infty}^{t} f_m(\tau)d\tau \tag{19}$$

Once the phase function is determined, we can obtain the signal for the $m$ instance of OFDM-NIFM as follows:

$$S_m(t) = \exp(j\theta_m(t)) \tag{20}$$

Combining the OFDM-NLFM φ-OTDR system with the basic system is essentially the same. The only difference lies in the driving signal generated by the arbitrary waveform generator. In the basic φ-OTDR system, the arbitrary waveform generator produces linear sweep pulses with exactly the same sweep range. Constrained by the length of the optical fiber, the time interval $T$ between each pulse must be greater than the total round-trip time of light in the fiber. However, in the combined OFDM-NLFM φ-OTDR system, the driving signal generated by the arbitrary waveform generator is as shown in Figure 3.

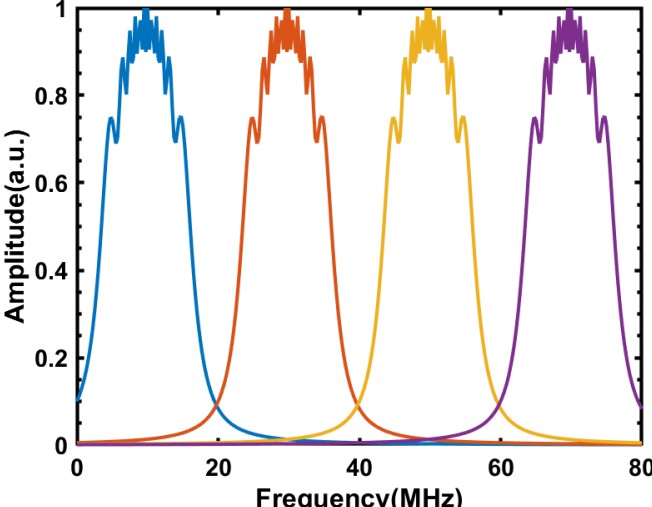

**Figure 3.** Spectrum of OFDM-NLFM signal.

The repetition period of the pulse combination remains $T$, but within each pulse combination, there are $N$ individual swept-frequency pulses with a time spacing of only $T/N$. By employing a frequency reuse algorithm, each pulse can effectively perform one detection cycle. This enhancement increases the system's sampling rate for vibrations by a factor of $N$ and also expands the vibration frequency response bandwidth by $N$ times.

Although the time interval between two pulses is merely $T/N$, when viewed in the frequency domain, the frequencies of adjacent pulses do not interfere with each other. Consequently, the Raman signals generated by these adjacent pulses in the frequency domain are also free from mutual interference, thus allowing them to be separated using a frequency-domain multiplexing algorithm.

The spectrum diagram of the OFDM-NLFM signal is shown in Figure 3. When $\Delta f = B_s$, the spectra of each signal are precisely separate and arranged sequentially, with each occupying a distinct channel. This separation significantly reduces inter-signal interference, simultaneously enhancing spectrum utilization and favoring matched filtering.

### 3. Experiments and Results

As shown in Figure 4, the experimental setup utilizes a narrow-linewidth laser (TeraXion, SFFL-N-34 (Quebec City, QC, Canada)) with a central wavelength of 1550.2387 nm and a linewidth of 1.035 KHz. The laser output is split into local light and signal light using a 10:90 coupler. The signal light is intensity-modulated into an OFDM-NLFM signal using an electro-optic modulator (EOM, JDSU, X5 (Singapore)). This signal consists of 4 subcarriers, each with an NLFM signal with a sweeping bandwidth of 20 MHz. An acousto-optic modulator (AOM, Gooch & Housego, T-M080 (Ilminster, UK)) is used for sideband suppression and a frequency shift of 80 MHz. The modulated pulse light is then amplified by an erbium-doped fiber amplifier (EDFA). It passes through a circulator and then enters a grating filter to filter out the amplifier's spontaneous emission noise. Subsequently, the light enters another circulator and travels through approximately 16.3 km of optical fiber. At around 14.3 km, the first piezoelectric transducer (PZT) with 5 m of wound optical fiber is positioned. Following this, it is connected to a 2 km-long optical fiber spool. A second PZT, which also has 5 m of wound optical fiber, is placed. Finally, there is a remaining 20 m of tail fiber at the end. The fiber generates backward Rayleigh scattering signals. These signals, along with the local light, are combined in the circulator. The beating frequency signal is detected using a balanced photodetector (BPD, MBD-200M-A) with a bandwidth of 200 MHz.

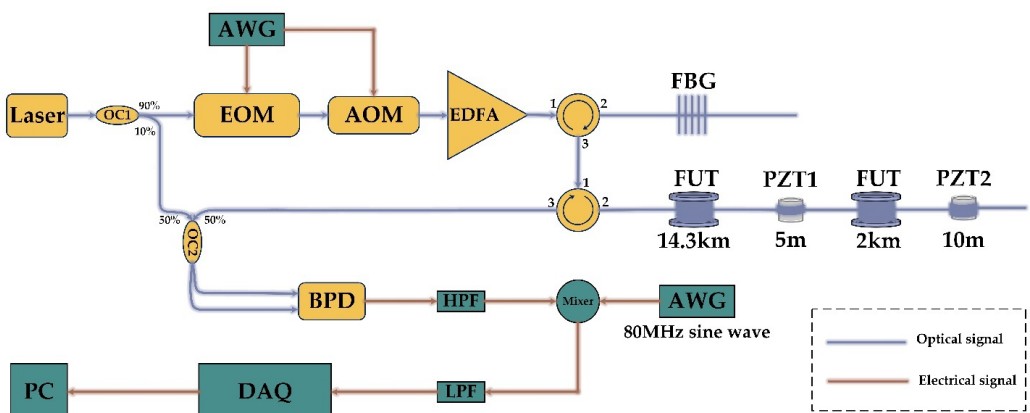

**Figure 4.** Experimental setup.

The converted electrical signal first passes through a high-pass filter with a cutoff frequency of 80 MHz. It is then mixed with an 80 MHz sine wave and subsequently goes through a low-pass filter with a cutoff frequency of 120 MHz. Afterward, the signal is sampled by an acquisition card, with a sampling rate of 250 MSa/s. The acquired data are processed using a PC, where each frequency band is down-converted separately,

followed by low-pass filtering and matched filtering. This process extracts the amplitude for positioning and simultaneously extracts the phase to reconstruct the vibration waveform.

When the subcarrier spacing is less than 1 times the sweep bandwidth, the received signal's positioning SNR and spatial resolution degrade. A 9.8 kHz 1 Vpp sine wave drive voltage is still applied to the PZT, with a sweep bandwidth of 20 MHz, theoretically achieving a spatial resolution of 10 m. As an example, considering a subcarrier spacing equal to 0.75 times the sweep bandwidth, the bandwidth occupied by the four frequency components should be $20 \times 0.75 \times 3 + 20 = 65$ MHz, as shown in Figure 5:

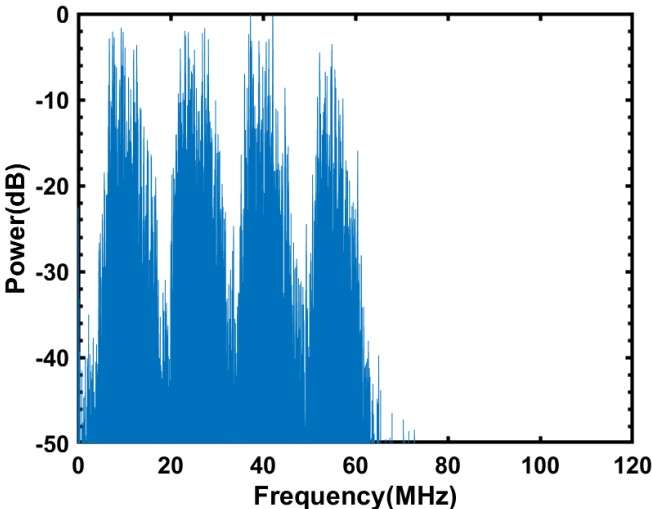

**Figure 5.** The power spectrum of the OFDM-NLFM pulse when the center frequencies of the subcarriers are spaced by 0.75 times the sweep bandwidth.

The dynamic range of the received signal after matched filtering is approximately 15 dB. The vibration position is determined using a moving differential method, and the result is shown in Figure 6a. A peak appears at 16.3 km in Figure 6a, indicating the location of the applied 9.8 kHz sine wave vibration. The SNR for this positioning is approximately 5 dB. As shown in Figure 6b, the moving differential peak of the tested optical fiber at 16.3 km is amplified. The indicated spatial resolution is approximately 11 m, which has decreased by about 1 m compared to the theoretical value.

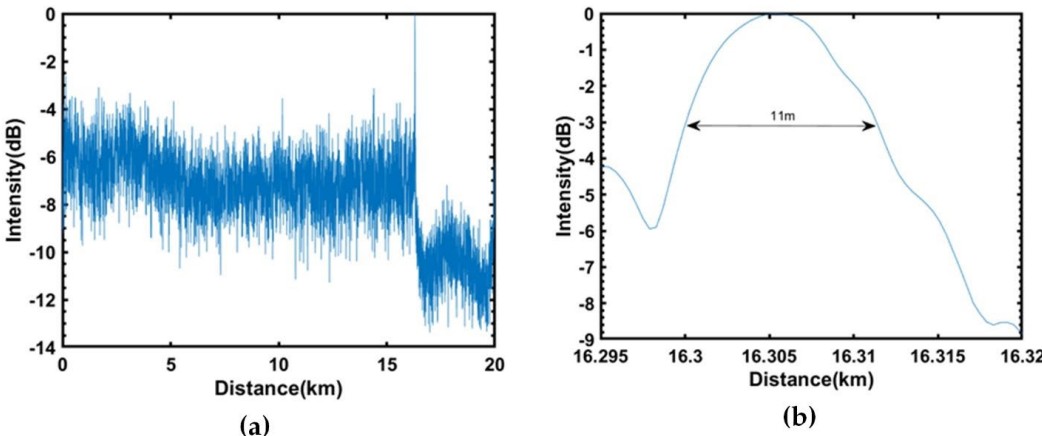

| (a) | (b) |
|:---:|:---:|

**Figure 6.** (**a**) The result of the moving differential after matched filtering of the received signal; (**b**) spatial resolution.

The curves 1, 2, 3, and 4 in Figure 7a represent the results of separate phase demodulating signals from four different frequency bands. Specifically, curve 1, which is represented

by the blue curve, shows the time-domain waveform of the signal in the 0–20 MHz frequency band after phase demodulation. Curve 2, which is represented by the red curve, represents the time-domain waveform of the signal in the 15–35 MHz frequency band after phase demodulation. Curve 3, which is represented by the yellow curve, displays the time-domain waveform of the signal in the 30–50 MHz frequency band after phase demodulation. Lastly, curve 4, which is represented by the purple curve, illustrates the time-domain waveform of the signal in the 45–65 MHz frequency band after phase demodulation. The results demonstrate sine waves with a frequency of 200 Hz in all cases, confirming the sensing system's ability to accurately capture all four frequency components of the vibration signal. The results shown in Figure 7a all indicate sine waves with a frequency of 200 Hz, confirming that this sensing system can accurately capture all four frequency components of the vibration signal.

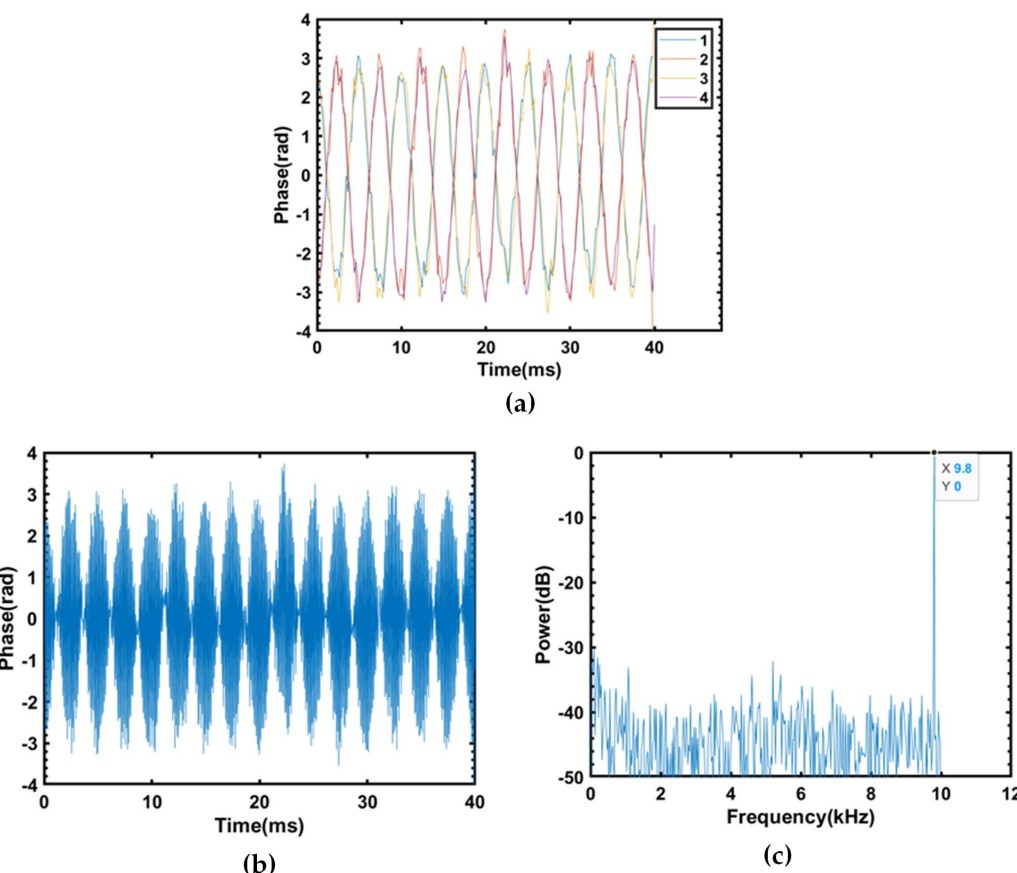

**Figure 7.** (**a**) The time-domain waveforms of the four vibration signals after phase recovery for the four carriers; (**b**) the time-domain waveform graph of the synthesized vibration signal from four signals; (**c**) the power spectrum of the synthesized vibration signal from four signals.

The amalgamated vibration waveform, which resulted from the sequential concatenation of phase-demodulated waveforms extracted from the 4 NLFM pulses, is visually represented in Figure 7b. Employing a fast Fourier transform (FFT) technique on the aforementioned vibration waveform, as showcased in Figure 7b, has enabled the derivation of the corresponding power spectrum, which is prominently depicted in Figure 7c. Within Figure 7c, a conspicuous peak is discernible at precisely 9.8 kHz, which is concomitant with the inherent frequency of the vibration signal, demonstrating an impressive SNR at approximately 40 dB. Figure 7b,c collectively underscore the capabilities of this sensing system, underpinned by its employment of OFDM. Notably, this system exhibits an unprecedented capacity for augmenting the frequency response range inherent to vibration signals. Moreover, the amalgamation of phase demodulation results from multiple

NLFM pulses validates its capacity to reconstruct high-frequency vibration waveforms with precision.

In the context in which the subcarrier center frequencies exhibit a spacing equivalent to 0.75 times the sweep bandwidth, an experiment was conducted involving the application of a swept sine wave signal with specific parameters. This signal entailed a rise time of 20 ms, a fall time of 20 ms, a frequency range spanning 1–10 kHz, and an amplitude characterized by a peak-to-peak driving voltage of 0.5 Vpp. The subsequent phase-recovered time-domain signal has been thoughtfully depicted in Figure 8a, while the outcome of subjecting the time-domain signal to short-time Fourier transform (STFT) is meticulously presented in Figure 8b. These visual representations, denoted as Figure 8a,b, serve as empirical evidence affirming the capability of this sensing system. Employing FDM, this system effectively expands its frequency response range from the initial 1~2.5 kHz to a more comprehensive 1~10 kHz range, marking a substantial fourfold enhancement in comparison to traditional single-frequency pulse approaches.

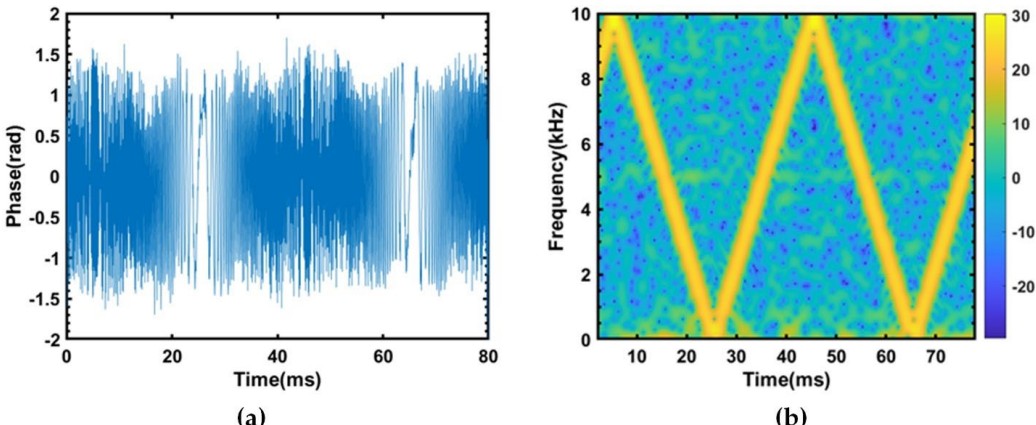

(a)  (b)

**Figure 8.** (**a**) Time-domain waveform of the vibration signal with a 1–10 kHz frequency sweep at a single position; (**b**) STFT (short-time Fourier transform) results of the vibration signal with a 1–10 kHz frequency sweep at a single position.

When the subcarrier center frequency spacing was 0.75 times the sweep bandwidth, a triangular wave with a frequency of 200 Hz and an amplitude of 5 Vpp was applied to PZT1, while a sine wave with a frequency of 4.8 kHz and an amplitude of 1 Vpp was applied to PZT2. The results of the motion differential are shown in Figure 9, with a localization SNR of approximately 8 dB.

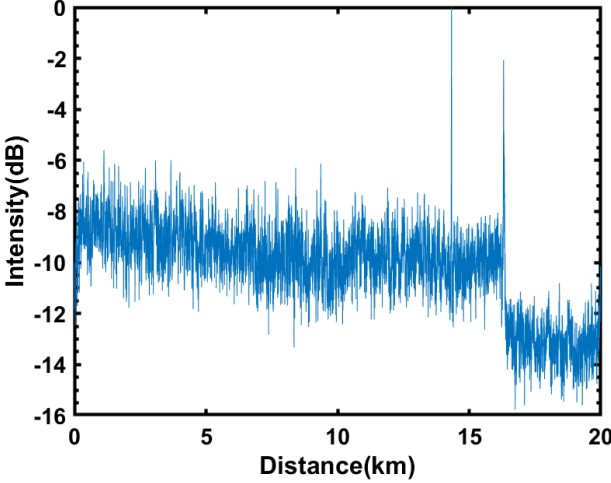

**Figure 9.** The results of motion differential for fixed-frequency vibration signals at multiple locations when the subcarrier center frequency spacing is 0.75 times the sweep bandwidth.

The vibration applied to PZT1 is depicted in Figure 10a in the time domain, with the corresponding power spectrum shown in Figure 10b, which exhibits an SNR of approximately 55 dB. For the vibration applied to PZT2, its phase-recovered time-domain signal is illustrated in Figure 10c, accompanied by the power spectrum shown in Figure 10d, which boasts an SNR of roughly 35 dB. These representations in Figure 10a–d collectively validate the system's capability to identify and accurately reconstruct the waveforms of multiple-point vibrations.

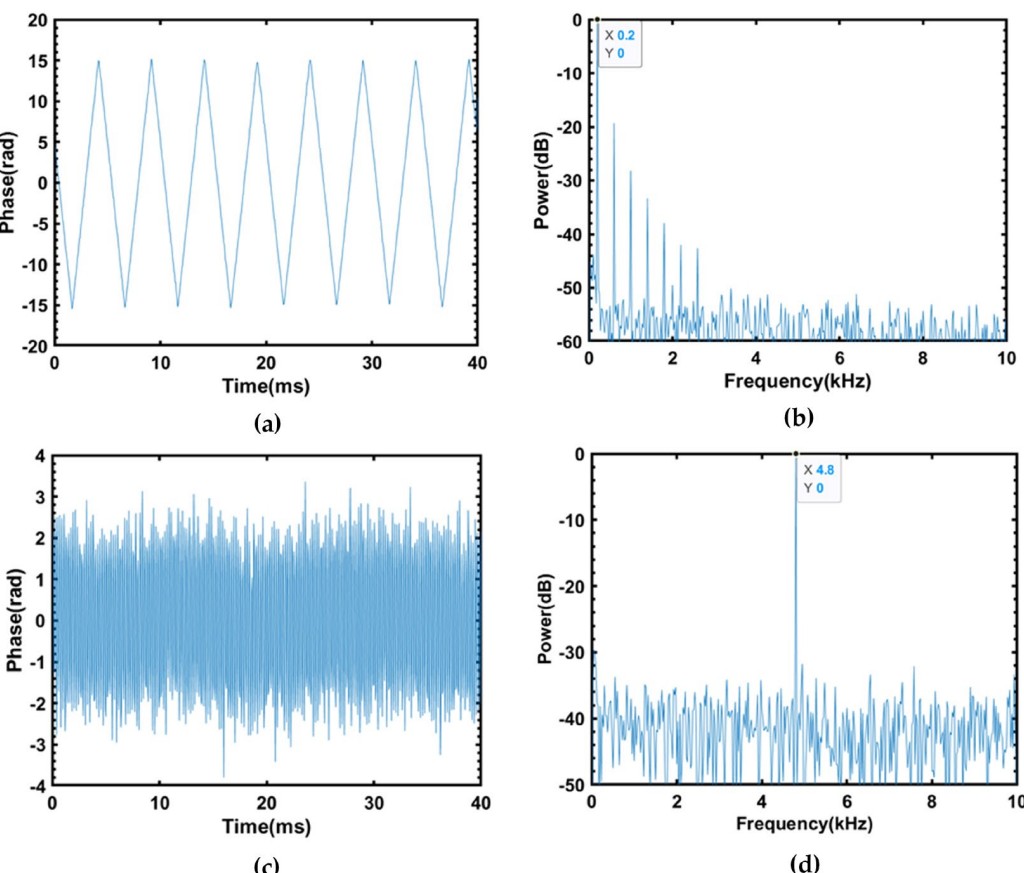

**Figure 10.** (**a**) The time-domain signal of the vibration applied to PZT1; (**b**) the corresponding phase-recovered frequency-domain signal; (**c**) the time-domain signal of the vibration applied to PZT2; (**d**) the corresponding phase-recovered frequency-domain signal.

## 4. Conclusions

This study presents a novel φ-OTDR technique that integrates OFDM and NLFM pulse modulation. The application of OFDM-NLFM signals addresses the challenges related to long-distance sensing, particularly in terms of limited frequency response bandwidth. By increasing the repetition frequency of the probing light through frequency multiplexing, this approach enhances spectral efficiency and expands the system's frequency response spectrum. Additionally, it enables more advanced frequency division techniques within the receiver's confined bandwidth. Fine-tuning the degree of nonlinearity in frequency modulation improves sidelobe suppression, reducing interference between adjacent signals and enhancing the system's SNR.

In summary, our investigation demonstrates optical fiber sensing with a 65 MHz frequency bandwidth, with multiplexing of four distinct frequency components using OFDM-NLFM signals. This innovative approach achieves an 11 m spatial resolution over a 16.3 km optical fiber span while covering a frequency response range from 1 to 10 kHz. These empirical findings highlight the substantial potential of this technique in extending the frequency response capabilities of φ-OTDR systems, paving the way for its integration in diverse

optical fiber sensing applications. Future studies may explore further advancements in signal processing and parameter optimization to maximize performance and applicability.

**Author Contributions:** All authors contributed substantially to the manuscript. Conceptualization, Z.L. and X.Y.; methodology, X.Y.; software, Y.Z. (Yuan Zhang); validation, Z.L., Z.X. and X.Y.; formal analysis, X.Y.; investigation, Z.L.; resources, Z.L.; data curation, Y.Z. (Yuan Zhang); writing—original draft preparation, Z.L.; writing—review and editing, Z.L.; visualization, Z.L.; supervision, Y.H., Y.Z. (Yangan Zhang) and X.Y.; project administration, Y.Z. (Yangan Zhang); funding acquisition, Y.Z. (Yangan Zhang) and X.Y. All authors have read and agreed to the published version of the manuscript.

**Funding:** This research was funded by State Key Laboratory of Information Photonics and Optical Communications (BUPT) (Grant No. IPOC2021ZT14), P. R. China; The Funds for Creative Research Groups of China (Grant No. 62021005).

**Institutional Review Board Statement:** Not applicable.

**Informed Consent Statement:** Not applicable.

**Data Availability Statement:** The data presented in this study are available on request from the corresponding author.

**Acknowledgments:** We thank the reviewers for their valuable feedback and the editors for their devoted guidance.

**Conflicts of Interest:** The authors declare no conflict of interest.

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
