# Peer review of "φ-OTDR Based on Orthogonal Frequency-Division Multiplexing Time Sequence Pulse Modulation"

_applsci, doi:10.3390/app132011355_

Round 1
Reviewer 1 Report
This paper proposed a phase sensitive OTDR technique using modified NLFM time sequence pulse modulation. The paper can be considered to be published after addressing the following issues/questions.
1) the conventional OFDM techniques are signal modulation and demodulation techniques used for data transmissions. While the proposed OTDR techniques cannot offer data transmission functionalities, using 'OFDM' is misleading. Considering the proposed technique is based on frequency division multiplexing NLFM pulses, it is suggested to use the term 'FDM' rather than 'OFDM'.
2) in the last paragraph of the Introduction, the authors have clarified that 'The combination of OFDM and NLFM in pulsed light generates .... in long-distance sensing scenarios.' The results presented in this paper, however, cannot fully verify such statements. It is recommended that the authors thoroughly compare the proposed techniques and the conventional OTDR techniques based on NLFM pulses using similar experimental setups.
3) to improve the readability of this paper, the authors should illustrate a clear transceiver digital signal processing procedure.
The paper should be carefully modified by the authors. Some obvious errors or typos would be corrected such as:
1) the x-axis title of Fig 2(a)
2) in the first paragraph of Section 3, it is mentioned that 'It passes through a circulator, a grating filter, and an amplifier spontaneous emission noise eliminator.' However, the 'amplifier spontaneous emission noise eliminator' is not plotted in Fig. 4. Or whether it is the grating filter.
Reviewer 2 Report
The authors, introduces an φ-OTDR technique that combines OFDM and NLFM time-series pulse modulation. It employs multiple orthogonal subcarriers to efficiently stack spectra, enhancing spectrum utilization and extending the system's frequency response range. Within the finite detector bandwidth, this approach allows for more frequency band multiplexing. NLFM technology is used to fine-tune frequency modulation nonlinearity, resulting in reduced side-lobe levels, improved side-lobe suppression ratios, reduced interference between neighboring signals, and an enhanced SNR. The combination of OFDM and NLFM in pulsed light generates wider pulse widths, enabling more optical energy injection into the sensing fiber and yielding higher backscattered optical power in long-distance sensing scenarios. I have a positive opinion about the manuscript and I find the manuscript is a good addition considering the scope of the Journal of Applied Sciences. I recommend this manuscript for publication after addressing the following concerns within minor revision.
- The title is clearly reflecting the content of the paper.
- Abstract: the abstract effectively summarizes the manuscript and well understood.
· Key words: were used properly.
- In introduction part: The reasons for performing this study need to be clarified with more details.
- Principle and theoretical analysis parts: this parts are well presented. However, all parameters that are used in all equations should be defined such as in eqn. 3 (K and k , are both for same parameter? If yes, then both should be written in same symbol).
- Results and Discussion parts: in fig. 7 (a) the colored lines 1 to 4 should be defined.
- The conclusion was clearly presented and the future work was suggested.
- The references, most of references are up to date.
Round 2
Reviewer 1 Report
The authors have addressed all my questions.